# Detection of Atrial Fibrillation Using 1D Convolutional Neural Network

**DOI:** 10.3390/s20072136

**Published:** 2020-04-10

**Authors:** Chaur-Heh Hsieh, Yan-Shuo Li, Bor-Jiunn Hwang, Ching-Hua Hsiao

**Affiliations:** 1College of Artificial Intelligence, Yango University, Fuzhou 350015, China; chxie@ygu.edu.cn; 2Department of Computer and Communication Engineering, Ming Chuan University, Taoyuan 333, Taiwan; 04160275@me.mcu.edu.tw (Y.-S.L.); 04160781@me.mcu.edu.tw (C.-H.H.)

**Keywords:** electrocardiogram (ECG), atrial fibrillation (AF), convolutional neural network (CNN), deep learning

## Abstract

The automatic detection of atrial fibrillation (AF) is crucial for its association with the risk of embolic stroke. Most of the existing AF detection methods usually convert 1D time-series electrocardiogram (ECG) signal into 2D spectrogram to train a complex AF detection system, which results in heavy training computation and high implementation cost. This paper proposes an AF detection method based on an end-to-end 1D convolutional neural network (CNN) architecture to raise the detection accuracy and reduce network complexity. By investigating the impact of major components of a convolutional block on detection accuracy and using grid search to obtain optimal hyperparameters of the CNN, we develop a simple, yet effective 1D CNN. Since the dataset provided by PhysioNet Challenge 2017 contains ECG recordings with different lengths, we also propose a length normalization algorithm to generate equal-length records to meet the requirement of CNN. Experimental results and analysis indicate that our method of 1D CNN achieves an average *F*_1_ score of 78.2%, which has better detection accuracy with lower network complexity, as compared with the existing deep learning-based methods.

## 1. Introduction

An arrhythmia is an irregular heart beating problem. A common type of arrhythmia is atrial fibrillation (AF) caused by abnormal sinus rhythm and irregular heartbeat which could lead to complications such as heart attack [1]. According to a report from the National Institute of Health, there are 2.2 million Americans suffering from AF and the possibility of having a stroke increases with age. A recent investigation [2] showed a 12% increase in the number of people in the UK diagnosed with heart failure from 2002 to 2014. These increasing heart disease problems lead to a heavy financial burden among countries. A study [3] showed that the average economic cost of heart failure was estimated at about USD 108 billion in 197 countries. Due to the increase of atrial fibrillation, development of an effective automatic detection method for AF from electrocardiogram (ECG) signals is valuable for healthcare. However, so far it is still a challenging task for computers due to its episodic nature [4,5].

ECG is a medical signal which tests the abnormality of the heart by measuring its electrical activity. A beat of ECG signals can be observed by five characteristic waves—P, Q, R, S, and T—as shown in Figure 1 [6]. P wave stands for atrial depolarization; QRS complex corresponds to the depolarization of the right and left ventricles, and contraction of the ventricular muscles; T wave represents repolarization of the ventricles.

The characteristics of AF can be diagnosed by ECG, including no clear P wave, abnormal pattern of R-R interval, irregularity of heart beating, and small oscillation (frequency 4–8 Hz) of ECG baseline.

The abnormality of the ECG signal is generally identified by using classification techniques [7], which are achieved by extracting the features of the ECG that can discriminate the defined categories. Conventionally, the features are derived from the magnitude, duration, and area of the QRS, T, and possibly P waves [8].

In order to improve detection and classification performance of ECG signals, several analysis methods based on conventional machine learning have been developed [8,9,10,11,12,13,14,15,16,17,18]. They first obtain hand-crafted features through conventional feature extraction and selection algorithms, and then train a classification model using these features. The typical features such as QRS complex, R-R interval, R amplitude difference, and heart rate turbulence offset, are extracted from ECG signals. Various classifiers have been applied to the problem including particle swarm optimization (PSO), support vector machines (SVM), self-organizing map (SOM), fuzzy c-means (FCM) clustering, rough sets, k-nearest neighbor (KNN), artificial neural network (ANN) classifier, and quantum neural network (QNN) [8,9,10,11,12,13,14,15,16,17,18,19]. 

The quality of extracted features significantly affects the performance of the detection/classification. However, the hand-crafted features are not generally robust with respect to many variations, such as scaling, noise, displacement, etc. Recently, deep learning (DL) has achieved great success in various problems, especially in computer vision and speech recognition. Several researchers have applied DL frameworks such as stacked auto-encoder, convolutional neural networks (CNNs), and long short-term memory (LSTM) for the analysis of ECG signals [20,21,22,23,24].

Luo et al. [20] presents a patient-specific deep neural network (DNN) heartbeat classifier. A stacked denoising auto-encoder (SDA) is pretrained by a time–frequency spectrogram obtained with unlabeled modified frequency slice wavelet transform. Then, a deep neural network model is initialized by the parameters of the trained SDA and then updated with fine-tuning.

In another study [21], the authors apply the approach of Hannun et al. [22] who constructed a 34-layer 1D ResNet for classification of the PhysioNet Challenge dataset. They also slightly modified the ResNet by reducing the number of filters per convolutional layer and using different input segment lengths. For simplicity we refer to the two methods as ResNet-1 and ResNet-2, respectively. 

Warrick et al. [23] proposed a CL3 (one CNN and three LSTMs) network for ECG classification that consists of two learning stages: representation learning and sequence learning. The former employs a 1D CNN to extract local and discriminative features of input signals; the latter is implemented with three stacked LSTM layers to learn long-term features.

Zihlmann and colleagues [24] proposed a method which applies the spectrogram of the ECG signal to train on a convolutional recurrent neural network (CRNN). This method first transforms ECG data into a 2D spectrogram using fast Fourier transform. The spectrogram is fed into a stack of 2D convolutional blocks and then a three-layer bidirectional LSTM network is followed for feature aggregation. However, the network complexity is rather high with more than 10 million training parameters.

In summary, the existing ML-based methods utilize expert knowledge to extract features, which is both time consuming and error-prone. The DL-based methods can be categorized into 1D and 2D schemes. In the 1D scheme, the time-series 1D data is directly fed into the neural network. On the contrary, for the 2D scheme, the time-series signal is converted into a 2D form (i.e., a spectrogram) before inputting the subsequent neural network. Compared to 1D schemes, 2D schemes improve the classification accuracy by around 3% [21,23,24] but at the cost of overhead of conversion and increasing the network complexity.

In order to raise the detection accuracy and reduce network complexity, this paper proposes an AF detection method based on an end-to-end 1D CNN architecture. By investigating the impact of major components of a convolutional block on detection accuracy and using grid search to obtain optimal hyperparameters of CNN, we develop a simple, yet effective 1D CNN. The 1D time-series raw data is fed into the network without any conversion. In addition, this AF detection is data driven, (i.e., the features of the ECG signal and classification model are learnt from input data directly). Since the dataset provided by PhysioNet Challenge 2017 contains ECG recordings with different lengths, we also propose a length normalization algorithm to generate equal-length records to meet the requirement of the CNN.

Conventionally, prediction accuracy (detection or classification accuracy) is often used as a metric for the evaluation of DNN-based classifiers. However, the complexity of DNNs is also important since it affects the computational load and implementation cost of a system. Thus, in this paper we use prediction accuracy and network complexity, presented in our previous work [25], as the metrics for the evaluation and comparison of the deep neural networks. The major contributions of this work are summarized as follows:We propose a simple, yet effective AF detection method based on end-to-end 1D CNN architecture to classify time-series 1D ECG signal directly without converting it to 2D data. Through exhaustive evaluation, we prove our method achieves better detection accuracy than the existing DL-based methods. In addition, the proposed method reduces network complexity significantly, as compared with the second-ranked method, CRNN.We study the effect of the batch normalization and pooling methods on detection accuracy, and then design the best network by combing the grid search method.We present a length normalization algorithm to solve variable length of ECG recordings.

The remainder of this paper is organized as follows. Section 2 first gives an overview of the proposed method, describes the data pre-processing algorithm, and then the design of 1D CNN. The numerical analysis and performance comparison are given in Section 3. Finally, the conclusion is drawn in Section 4.

## 2. Proposed AF Detection

### 2.1. System Overview

The proposed AF detection system shown in Figure 2 includes data length normalization, offline training, and online prediction. The length of ECG recordings of the dataset we used is variable, and thus not suitable for the deep neural network. Therefore, we developed a pre-processing algorithm to make each recording fixed in length. The dataset contains four classes of ECG signals: AF, Normal, Noisy, and Other. The detection of AF is easily formulated as a classification problem. Thus, we designed a 1D CNN for the classification of the ECG signals. The classifier design consists of training and inference. In the training phase, the 1D CNN predicts one of the four classes for training data. The loss (error) between the predicted output and ground truth output is calculated and then back-propagates to each layer of the network so as to update the parameters of the network iteratively. An optimal network model is obtained when the iteration process converges. In the inference phase, the test data is applied to the 1D CNN with the optimal model, and the predicted result is generated.

### 2.2. Data Length Normalization

Due to the difficulty of putting variable-length data into a deep neural network, length normalization, which makes each recording into a segment with a fixed length, should be performed first. The length normalization problem is rather common across many domains. In studies by Andreotti et al. [21] and Zhou et al. [26], the authors manually preset a required length (called length threshold here), and then cut the original data into segments with the required length. The method is rather simple, but its major problem is that the setting of the length threshold is done manually, which does not consider the characteristic of the dataset. Setting length threshold to any value in the range of the recording lengths (9 s to 61 s) will generate bias. The problem is how to minimize the bias by generating an appropriate length threshold in an automatic mechanism.

In this work, we develop a histogram-based length normalization algorithm, which automatically determines the length threshold value using the histogram distribution of the length of the recordings. The length threshold calculated by this algorithm is 30 s, which corresponds to 9000 samples since the sampling rate is 300 samples per second. The histogram of the lengths of 8528 recordings is shown in Figure 3. It indicates that the length of the majority of data falls in the range of 30 s. Obviously, our algorithm can adapt the characteristic changes of input data and will create the least bias in the statistical sense since it considers the most frequent data length as a threshold.

The normalization algorithm not only solves the variable length data problem but also generates more data to train the deep neural network. The pseudocode of the algorithm is shown in Algorithm 1. If the recording contains just 9000 samples (30 s), this recording will be used directly without any modification. If the recording contains more than 9000 samples, multiple segments will be generated with 50% overlap between adjacent segments. Each segment from the same recording has 9000 samples and will be assigned the same label. For instance, an AF recording containing 18,600 samples (61 s) will be replaced by four segments containing 9000 samples and labeled as AF. Finally, if the length is less than 9000, we concatenate the recording with the same label to reach 9000 samples. After pre-processing, the number of recordings for AF, Normal, Noisy, and Other is 903, 5959, 299, and 2990, respectively, increasing from 8528 to 10,151 in total.
**Algorithm 1.** Pseudocode of data length normalization.1: IF the length of the recording is greater than 9000 samples2: Chop recording into 9000 samples with 50% overlap between segments3: IF the length of the recording is less than 9000 samples4: DATA: = copy the recording5: Append DATA in the back of the recording6: DO step 5 until the appended recording reaches 9000 samples7: IF the length of the recording is equal to 9000 samples8: Preserve the recording

### 2.3. 1D CNN Design

#### 2.3.1. CNN Architecture

The design of deep neural network architecture is rather challenging, and involves several aspects such as performance metric, loss function, optimization algorithm, and hyperparameter setting [27]. The hyperparameter setting includes the settings of the number of hidden layers, number of neurons and number of channels for every layer, learning rate, batch size, batch normalization, pooling, and so forth. Transfer learning has been widely used in 2D images. Several excellent pre-trained 2D networks such as LeNet, AlexNet, VGG, Inception, and ResNet can be applied to other image processing tasks just with simple fine-tuning [28,29]. However, there are very few pre-trained networks in one dimension, hence we designed the network from scratch. Based on our experiences and utilizing the grid search [27], we obtained a 10-layer architecture through the process of trial and error.

The proposed architecture contains 10 convolutional blocks, 2 fully-connected layers, and a Softmax layer as the output prediction, as shown in Table 1. The convolutional block consists of a convolutional layer, a Rectified Linear Unit (ReLU) layer, and a Maxpooling layer. We added a batch normalization (BN) layer [30] after ReLU activation only in the first convolutional block to normalize the input layer by adjusting and scaling the activations. BN normalizes the input to zero mean and unit variance, which improves the performance and stability of the deep neural network [27]. Five convolutional blocks are then followed with a dropout layer. The next convolutional blocks share the same structure and then apply a dropout layer. In the last block, we only applied a convolutional layer and a ReLU layer. The arrangements of batch normalization and pooling methods affect the prediction accuracy, which will be discussed in the following section.

As shown in Table 1, the filter size of the convolutional layer starts with 32, meaning 32 kernels will convolve with input data and output 32 feature maps. The filter size is increased by multiplication of 2 every 2 blocks in order to retrieve more combinatory features. The kernel size in all layers is set to five for the reduction of computational load. 

We downsampled each convolutional output by means of a pooling layer with a kernel size of two and used the Softmax function in the last layer to produce prediction probability over the four output classes. The Softmax function, defined in Equation (1), maps the real-valued input into prediction probability. Ground truth labels are transformed into one-hot encoding vector. In our case, we have four labels: AF, Normal, Noisy, and Other. Each label is a vector, containing either 1 or 0. If the label is “AF,” the one-hot encoding vector will be [1, 0, 0, 0], giving the probability of each label.
(1)$$\mathrm{We}Softmax(z_{j})=\frac{e^{z_{j}}}{\sum_{k=1}^{K}e^{z_{k}}},\;j=1,\;\ldots K,$$

#### 2.3.2. CNN Learning

The CNN model is obtained by training with the adaptive moment estimation (Adam) [31], which is a variant of the stochastic gradient descent (SGD) optimization algorithm. The Adam utilizes the backpropagation (BP) method to learn the parameter set of the neural network by minimizing the loss function which corresponds to the difference between network prediction and the ground truth. 

In this CNN, we choose the cross-entropy as loss function, *J*(***x***, ***y***, *θ*) defined in Equation (2). The learning of the network is to minimize *J*(***x***, ***y***, *θ*) with respect to network parameter set *θ*.
(2)$$J(x,y,\theta )=-\sum_{i=1}^{K}x_{i}logy_{i},$$
where ***x*** and ***y*** denote the ground truth and the predicted output of the CNN, respectively, and *K* is the minibatch size.

In backpropagation training, the parameter set *θ* of the CNN is obtained by an iterative update as [32]:(3)$$\theta_{t}\leftarrow \theta_{t-1}-\frac{\eta {\hat{m}}_{t}}{\sqrt{{\hat{v}}_{t}}+\varepsilon },$$
where *η* is a fixed learning rate; *ε* is a very small constant; ${\hat{m}}_{t}$ and ${\hat{v}}_{t}$ are the bias-corrected first moment estimate and the biased-corrected second moment estimate, respectively, which are defined elsewhere [32]. 

Adam is an adaptive learning rate method, which computes individual learning rates for different parameters. The default learning rate is 0.001 [32]. Generally, a large learning rate will make a model learn faster while a small learning rate will maintain better stability in the learning process. In the 1D CNN, the dropout layer is also employed between the two convolutional blocks to avoid overfitting. The dropout layer will randomly choose a percentage of neurons and update only the weights of the remaining neurons during training. [27]. We set the dropout parameter to be 0.5, which means half of the neurons will not be updated.

## 3. Numerical Analysis

### 3.1. Dataset

The dataset we used is provided by PhysioNet Challenge 2017, containing 8528 single lead variable-length ECG recordings lasting from 9 s to 61 s. The recordings are sampled at 300 Hz and have been bandpass filtered by the AliveCor KardiaMobile device, which generates lead I (LA-RA, i.e., when electrodes are placed on the subject’s left arm and right arm) equivalent ECG recordings [7]. ECG recordings are transmitted over the air using 19 kHz carrier frequency, digitized at 44.1 kHz and 24-bit resolution, and stored as 300 Hz and 16-bit files with a bandwidth of 0.5–40 Hz and a ±5 mV dynamic range [7]. The dataset is composed of four classes, including 771 AF (atrial fibrillation), 5154 Normal (normal sinus rhythm), 46 Noisy (too noisy to be recognized), and 2557 Other (other rhythm), which were annotated by experienced experts. The waveforms of examples of the four classes are shown in Figure 4. The “Other” represents recordings that can be clearly classified but do not belong to any of the other three classes (i.e., AF, Normal or Noisy). After data length normalization, the number of recordings becomes 10,151. The increase of data samples is beneficial for training in avoidance of overfitting.

### 3.2. Evaluation Metrics

As stated before, the proposed system predicts the input ECG record as one of the four classes. We adopt *F*_1_ score to evaluate the prediction (detection) performance, which is calculated as
(4)$$F_{1}\left(\%\right)=\left(\frac{2*\mathrm{Precision}*\mathrm{Recall}}{\mathrm{Precision}+\mathrm{Recall}}\right)\;*\;100,$$

Precision and recall metrics can be derived from the confusion matrix, which shows the performance of a classification algorithm. Each row of the matrix stands for a predicted class while each column represents a ground truth class. The subscript indicates the number of classifications. For example, A represents AF as ground truth but predicted noisy as the outcome. A confusion matrix of four classes [33] is shown in Table 2.

To calculate *F*_1_ of each class, we transform the confusion matrix of size 4 × 4 into that of size 2 × 2. Each row and column contain two classes: target label and remaining label. A 2 × 2 confusion matrix of AF is shown in Table 3. “Non-AF” includes Normal, Noisy, and Other classes.

Precision, recall, and accuracy of AF are calculated as follows:(5)$$\mathrm{Precision}=\frac{A_{a}}{A_{a}+NA_{a}},$$
(6)$$\mathrm{Recall}=\frac{A_{a}}{A_{a}+A_{na}},$$
(7)$$\mathrm{Accuracy}=\frac{A_{a}+NA_{na}}{A_{a}+NA_{a}+A_{na}+NA_{na}},$$

Accuracy is used in most cases in terms of evaluating how good the classification is. However, accuracy is inadequate for an unbalanced dataset due to the majority of negative class. Therefore, we use *F*_1_ score as an alternative measure since it is a balance of precision and recall metrics and is useful when dealing with uneven class distribution [32].

*F*_1_ scores for the other classes (i.e., Normal (*F*_1*N*_), Noise (*F*_1~_), and Other (*F*_1*O*_)), are calculated in the same way as AF (*F*_1*A*_). The overall detection performance of the proposed method is evaluated by the average of *F*_1_ scores of the four classes [33] as follows:(8)$$\mathrm{Average}\;F_{1}=\frac{F_{1N}+F_{1\sim }+F_{1A}+F_{1O}}{4},$$

### 3.3. K-Fold Cross-Validation

K-fold cross-validation (K-fold CV) is the most popular method in various applications of machine learning [27,34,35]. Stratified K-fold CV is a stratified-sampling version of K-fold which returns stratified folds: each set contains approximately the same percentage of samples of each target class as the complete set. It is superior to regular K-fold CV, especially for imbalanced classifications. As mentioned in the previous section, the sample distribution of the four classes in the pre-processed dataset which contains 10,151 segments (AF = 903, Normal = 5959, Noisy, = 299 and Other = 2990) is imbalanced. Therefore, we utilize stratified K-fold CV rather than regular K-fold CV.

In order to find the best K value, we divided the pre-processed dataset into training and test subsets with different size ratios, and then did training and testing for each size ratio. The results are shown in Table 4. It indicates that the train/test ratio of 80:20 achieves the best detection performance, which corresponds to K = 5. Therefore, we used a stratified five-fold CV in our work. The average *F*_1_ score of a class is calculated by averaging *F*_1_ of all folds, and the results are 79.1%, 90.7%, 65.3%, and 76% for AF, Normal, Noisy, and Other, respectively.

### 3.4. Hyperparameter Optimization

We performed a grid search algorithm over number of layers, kernel size, batch size, and learning rate, for hyperparameter optimization in training. Given a set of parameters for training, grid search yields an optimal model which maximizes accuracy on independent datasets. The suggested learning rate of 0.001 in Adam optimizer [31] is used as a base. In order to find the best accuracy around the base learning rate, we chose one parameter greater than the base and three more less than the base to form a set for learning rate as Lr ∈ {0.00005, 0.0001, 0.0005, 0.001, 0.005}. We selected five elements to form a batch-size set which started from 30 and increased by 20; i.e., Bs ∈ {30, 50, 70, 90, 110}. Three kinds of kernel sizes were selected as Ks ∈ {3, 5, 7}. As for number of layers in our case, the upper limit was 12 since we decreased the dimensionality of feature map after every convolutional layer. We define the set of number of layers as N ∈ {8, 9, 10, 11}. The combination of the above parameter sets yields 300 neural network architectures. We applied the grid search to obtain an optimal architecture and used four Graphics Processing Units (GPUs) (Nvidia GTX 1080 Ti) to speed up training.

### 3.5. Results and Analysis

#### 3.5.1. Prediction Accuracy

Each recording is assigned a label as AF, Normal, Noisy, or Other. The proposed 1D CNN model will infer the label on the test set. Using the grid search, we can obtain the best model and evaluate the average *F*_1_ score, which is the average of the four classes.

Applying the grid search to the dataset, we obtained the results demonstrated in Table 5. The table shows the best average *F*_1_ score and standard deviation σ under various combinations of network hyperparameters including number of convolutional layers, filter kernel size, batch size, and learning rate. For instance, for the CNN with eight convolutional layers, the first row of the table lists the best *F*_1_ scores of kernel sizes of 3, 5, and 7 with corresponding optimal batch sizes of 50, 50, and 70 and learning rates of 0.001, 0.001, and 0.0005, respectively. It is noted that for 11 convolutional layers, there is no output when kernel size is greater than 3. This is because there was no sufficient data to perform convolution in the last convolutional layer since the feature maps are downsampled using pooling at every convolutional layer. This table also shows the total number of network parameters for each architecture combination.

Table 5 indicates that the best combination is: number of layers = 10, kernel size = 5, learning rate = 0.0001, batch size = 30. The combination reaches average *F*_1_ score of 77.8%. Since the batch size of 30 is the searching lower bound, we further ran a small grid search with batch sizes of 10 and 20. The results show that the average *F*_1_ score of batch size of 10 and 20 is 76.4 and 77.1, respectively. This proves batch size of 30 is the best parameter.

Figure 5 shows the normalized average confusion matrix of five folds. The diagonal shows the number of correctly predicted records for each class. The off-diagonals display the number of misclassifications for each class. It is seen that 21 AF records are misclassified as other. This is because some AF records contain characteristics of the class “Other.” An example of an AF record which is misclassified as other is shown in Figure 6. In this figure, the orange rectangle shows that the ECG waveform has AF characteristics, which are affected by the abnormal mode of the R-R interval, while the yellow rectangle indicates that the ECG waveform does not have the characteristics of AF, Noisy, or Normal. 

Figure 5 also indicates imbalanced dataset problem. For the test set, the total number of records of the Normal class is 1215, and the total number of the records of the remaining classes is only 816. In other words, the Normal class dominates the training, and it is possible the learned model will tilt to the normal class.

#### 3.5.2. Network Architecture Analysis

Batch normalization is a renowned regularization technique that accelerates training [30] and is widely used in modern network architecture [22]. Maxpooling is a strategy that downsamples feature maps and is mostly placed after the convolutional layer for feature reduction [36]. Although these strategies proven to be robust and useful in several application domains, we conducted an experiment to look into the performance changes under various arrangements of batch normalization and Maxpooling. The proposed 1D CNN architecture shown in Table 1 contains only one batch normalization layer placed after the first convolutional layer, and one Maxpooling layer for every convolutional layer, except the last convolution layer. For convenience, the architecture is denoted as Proposed-1. Moreover, we replace every Maxpooling layer of Proposed-1 with average pooling, and denote this as Proposed-2. 

We also designed four variants of Proposed-1 and one variant of Proposed-2 as follows. The variants of Proposed-1 include (a) All-BN: add one batch normalization layer after every convolutional layer; (b) No-BN: no batch normalization layer in the network; (c) Maxpooling: add the Proposed-1 with one Maxpooling layer before the flatten layer; (d) Max-average pooling: add Proposed-1 with one average pooling layer before the flatten layer, which combines Maxpooling and average pooling. The variant of Proposed-2 is to add one average pooling layer before the flatten layer into the Proposed-2 network, which is denoted as Extra-Average.

Table 6 shows the average *F*_1_ scores of the four labels for the two proposed networks and their variants. The detection performance of All-BN degrades significantly, as compared to No-BN or our Proposed-1 architecture. This is due to gradient explosion [37] in deep layer that makes the network hard to train. Furthermore, batch normalization leads to instability of training [37]. Training/validation accuracy of the models All-BN and No-BN are depicted in Figure 7 and Figure 8, respectively. It is seen that the accuracy curve on the validation set fluctuates drastically for All-BN (Figure 7) while the accuracy curve for No-BN (Figure 8) is relatively smooth.

A vanilla CNN usually combines the convolutional layer and Maxpooling layer as a block and repeats this before the flatten layer [36]. We investigated how the Maxpooling layer placed before the flatten layer affects accuracy. The result shown as “Maxpooling” in Table 6 reflects that the accuracy degrades significantly, and this variant yields the worst accuracy. The reason can be traced back to the characteristics of Maxpooling. In Maxpooling operation, it only keeps the maximum element and removes the others. This method may not preserve the originality of the data and is likely to eliminate the distinguishing features in the same pooling region [38]. 

In order to investigate the effect of pooling methods, we replace Maxpooling in Proposed-1 with average pooling, and the result is shown as Proposed-2 in Table 6. It is obvious that Proposed-2 outperforms Proposed-1 and its variants. This may be due to the effect that average pooling keeps most information from previous layer and is able to pass this down layer by layer. In addition, the performance of Proposed-2 slightly degrades if we add one average pooling layer before flatten layer (Extra-Average in Table 6). However, the *F*_1_ score drops by 10% if we add one Maxpooling layer before the flatten layer in Proposed-1. The experiment shows that using average pooling is more stable than using Maxpooling.

#### 3.5.3. Network Complexity Analysis

To evaluate the network complexity, we estimated the total number of network training parameters using the deep learning platform Keras. The parameters of each layer, excluding non-trainable layers such as the pooling layer, dropout, and flatten layer, are shown in Table 7. The total number of network training parameters of the proposed 1D CNN is approximately 3 million.

For comparison, we also built up the CRNN using Keras and calculated the total number of network training parameters of the CRNN. The CRNN architecture mainly consists of two networks: 2D CNN and LSTM. The 2D CNN uses four convolutional blocks. Each block is made up of six 2D convolutional layers, followed by batch normalization and ReLU activation. The last layer of a block applies Maxpooling with a kernel size of 2. Feature maps from the Maxpooling layer are flattened before feeding into a three-layer bidirectional LSTM network. The total number of training parameters of CRNN is 10,149,440, which is three times more than ours and requires huge computation in training. Note that the transformation of 1D ECG signal to 2D spectrogram is not taken into account. In addition, 2D convolution of 24 layers and three-layer LSTM dominate the network complexity of the CRNN.

#### 3.5.4. Comparison of Various Methods

To prove the effectiveness of our AF detection system, we compared it with the existing DL-based methods [21,23,24]. The DL-based methods include 1D schemes: ResNet-1, ResNet-2, and CL3-I (CL3 experiment I [23]), and 2D scheme: CRNN, as mentioned in Section 1. The comparison metrics include detection accuracy of cross-validation and total number of network training parameters.

A detection accuracy comparison of our model with the existing DL-based methods for each class is summarized in Table 8. It indicates that our proposed method achieves better prediction accuracy for all classes. In addition, the detection performance of AF of this method is more than 4% higher than the second best CRNN. 

Table 9 shows the comparison using the metrics of average *F*_1_ score of four classes and average *F*_1_ score of three classes (excluding Noisy), and the total number of parameters. The results indicate that the proposed networks achieve an average *F*_1_ score higher than the existing networks. The existing 1D schemes, ResNet-1, ResNet-2 and CL3-I, perform much worse than our networks and CRNN in average *F*_1_ score of four classes. In addition, our network outperforms the CRNN with much lower network complexity (less than 1/3). This implies that both training cost and implementation cost of the proposed deep neural network are significantly lower than those of the CRNN.

## 4. Conclusions

This paper developed an end-to-end 1D CNN for the AF detection from time-series ECG data. By studying the impact of the BN and pooling methods on detection accuracy and combing the grid search method to obtain optimal hyperparameters, we designed a simple, yet effective 1D CNN network. We also developed a length normalization algorithm, which automatically determines the length threshold value. The algorithm made variable-length ECG records, fixed-length ones, and augmented training data. The algorithm adapted the characteristic change of data by using the histogram information of the input data.

In the literatre, BN is widely used to improve performance. However, in our application, when the BN layer was added subsequently to every convolutional layer (All-BN), F_1_ score dropped 7%, as compared to the case without BN (No-BN). When the BN layer was added only subsequently to the first convolutional layer (our Proposed-1 architecture), F_1_ score was further improved by 1.6%, as compared to No-BN. Although BN was able to speed up learning, our experiment indicated that it could lead to instability in training. As for pooling methods, we found that average pooling is able to keep information from the previous layer and pass to a latter layer, while Maxpooling only keeps the maximum value, which may lose important information sometimes. The experimental results showed that average pooling achieved a slightly higher F_1_ score than Maxpooling, and it was more stable in training.

Through exhaustive evaluation, we proved our 1D CNN method achieved better cross-validation detection accuracy than the existing methods. In addition, the proposed method reduced network complexity significantly, as compared with the second-ranked method, CRNN. Applying spatial pyramid pooling [39], which does not require fixed input size, in replacement of the flatten layer, and investigating a better way to solve the problem of extremely imbalanced dataset will be the future directions.

## Figures and Tables

**Figure 1 sensors-20-02136-f001:**
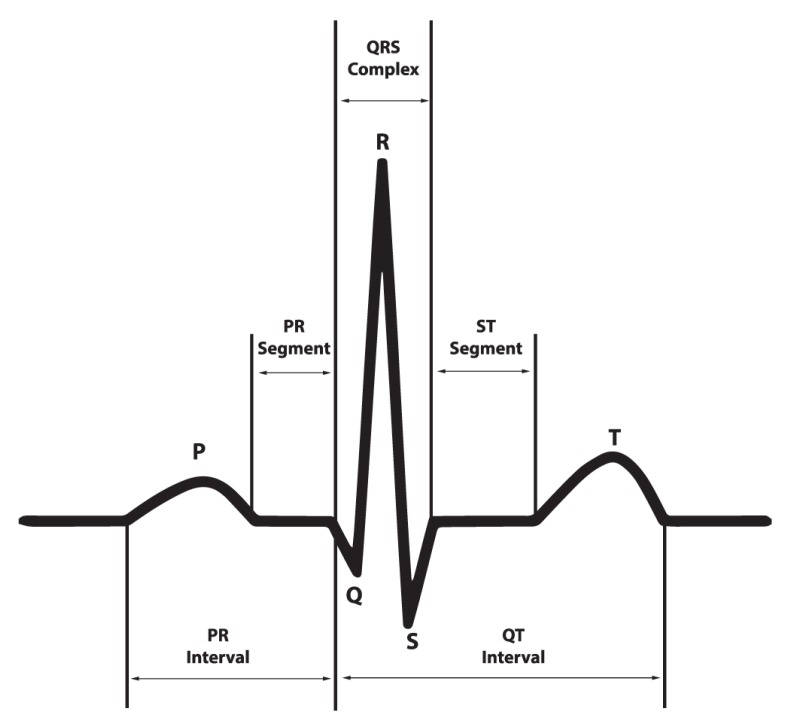
Electrocardiogram.

**Figure 2 sensors-20-02136-f002:**
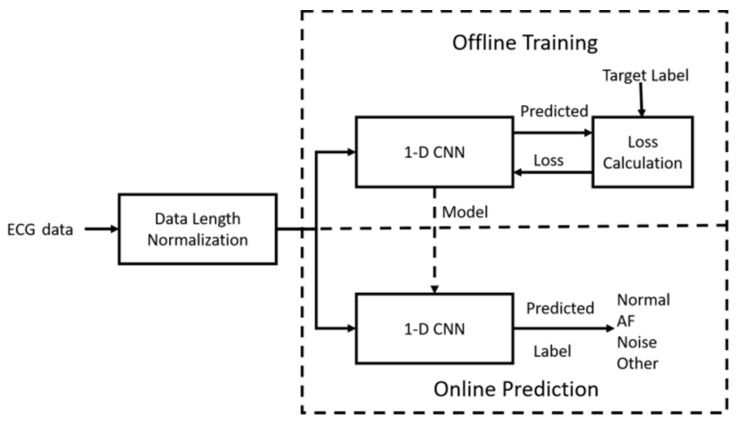
Flowchart of the proposed atrial fibrillation (AF) detection method. ECG: electrocardiogram; CNN: convolutional neural network.

**Figure 3 sensors-20-02136-f003:**
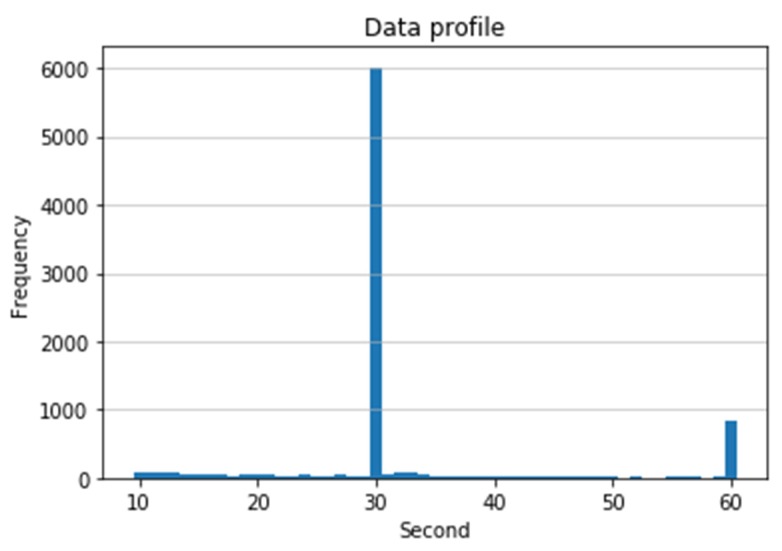
Data length histogram distribution.

**Figure 4 sensors-20-02136-f004:**
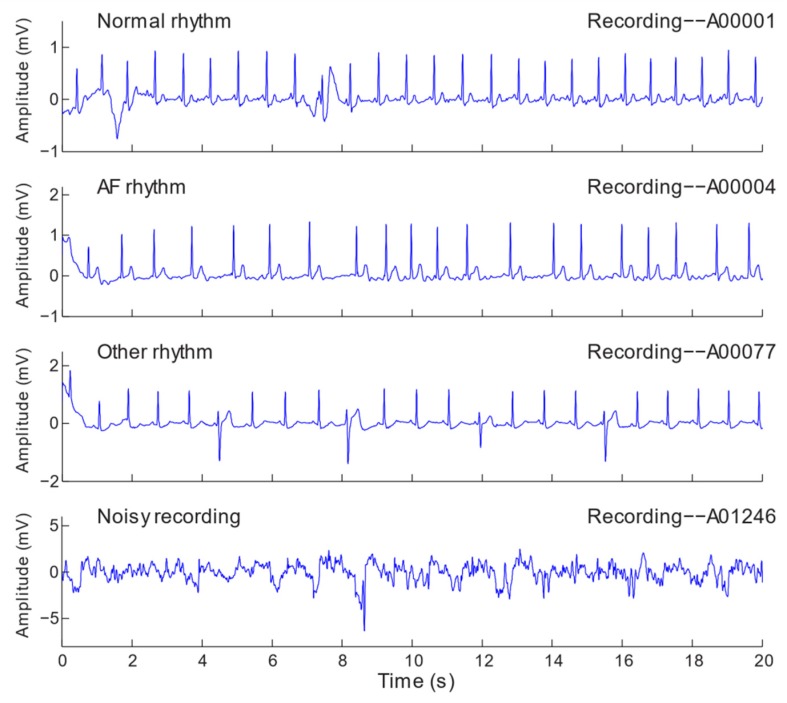
ECG examples of four classes: Normal, AF, Other, and Noisy.

**Figure 5 sensors-20-02136-f005:**
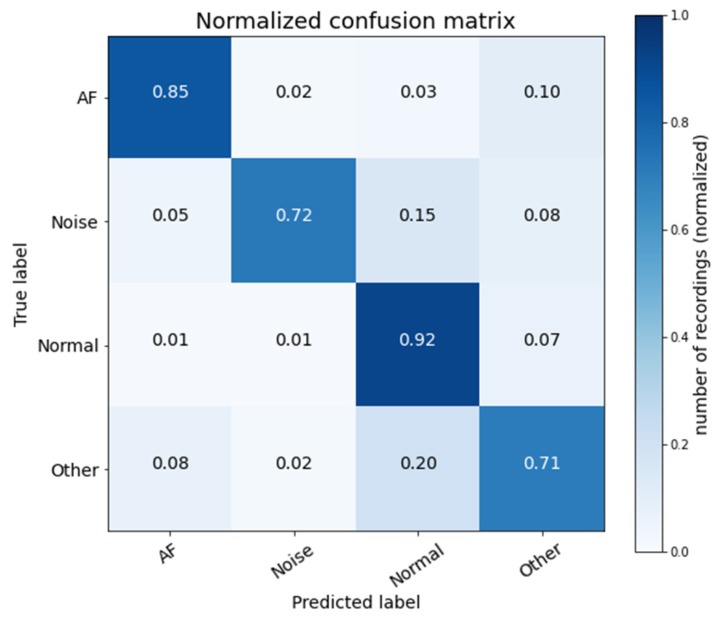
Normalized average confusion matrix of five folds.

**Figure 6 sensors-20-02136-f006:**
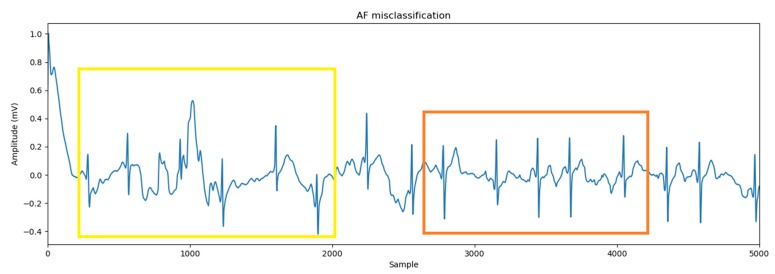
AF record which is misclassified as other.

**Figure 7 sensors-20-02136-f007:**
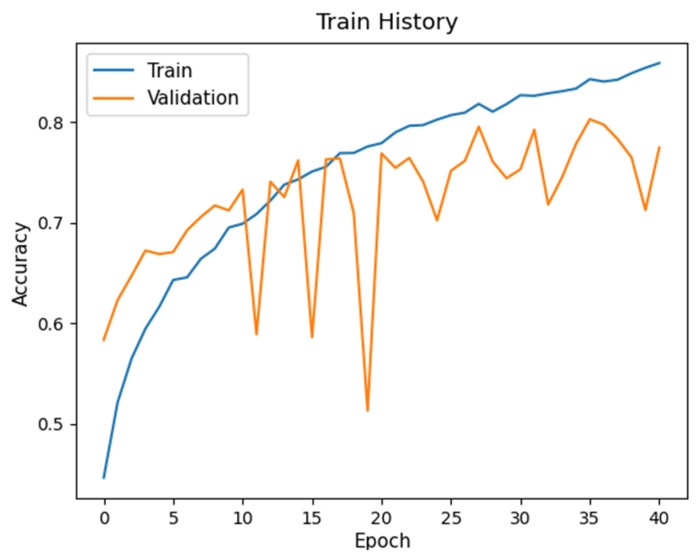
Training accuracy of All-BN.

**Figure 8 sensors-20-02136-f008:**
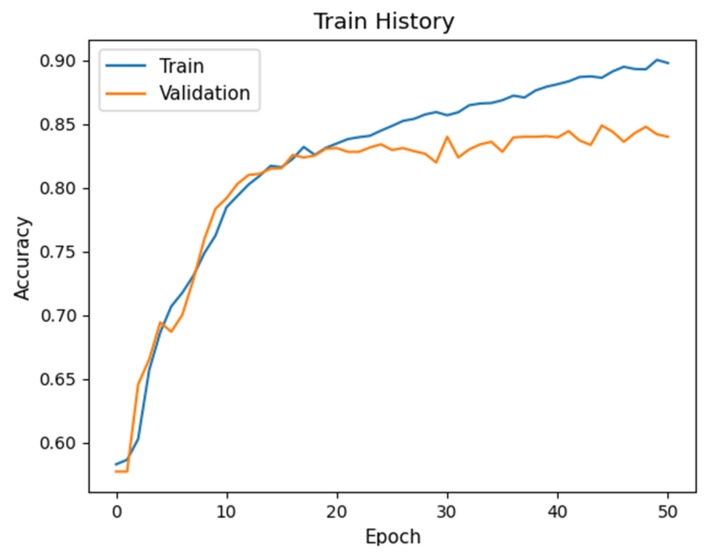
Training accuracy of No-BN.

**Table 1 sensors-20-02136-t001:** Parameters of each layer of proposed 1D convolutional neural network (CNN).

Layers	Parameters	Activation
Conv1D	Filter 32/kernel 5	ReLU
BN		
Maxpooling	2	
Conv1D	Filter 32/kernel 5	ReLU
Maxpooling	2	
Conv1D	Filter 64/kernel 5	ReLU
Maxpooling	2	
Conv1D	Filter 64/kernel 5	ReLU
Maxpooling	2	
Conv1D	Filter 128/kernel 5	ReLU
Maxpooling	2	
Conv1D	Filter 128/kernel 5	ReLU
Maxpooling	2	
Dropout	0.5	
Conv1D	Filter 256/kernel 5	ReLU
Maxpooling	2	
Conv1D	Filter 256/kernel 5	ReLU
Maxpooling	2	
Dropout	0.5	
Conv1D	Filter 512/kernel 5	ReLU
Maxpooling	2	
Dropout	0.5	
Conv1D	Filter 512/kernel 5	ReLU
Flatten		
Dense	128	ReLU
Dropout	0.5	
Dense	32	ReLU
Dense	4	Softmax

**Table 2 sensors-20-02136-t002:** Confusion matrix of four classes.

		Ground Truth			
		AF (A)	Normal (N)	Noisy (~)	Other (O)
**Predicted**	AF (a)	*A_a_*	*N_a_*	*~_a_*	*O_a_*
	Normal (n)	*A_n_*	*N_n_*	*~_n_*	*O_n_*
	Noisy (_~_)	*A_~_*	*N_~_*	*~_~_*	*O_~_*
	Other (o)	*A_o_*	*N_o_*	*~_o_*	*O_o_*

**Table 3 sensors-20-02136-t003:** Confusion matrix of two classes.

		Ground Truth	
		AF (A)	Non-AF (NA)
**Predicted**	AF (a)	*A_a_*	*NA_a_*
	Non-AF (na)	*A_na_*	*NA_na_*

**Table 4 sensors-20-02136-t004:** *F*_1_ scores using different train/test ratio.

Train/Test	AF	Normal	Noisy	Other	Average
60:40	72.0	90.0	60.0	69.0	72.75
70:30	76.0	90.0	60.0	70.0	74.00
80:20	77.0	89.0	67.0	74.0	76.75
90:10	76.9	90.0	62.1	72.8	75.45

**Table 5 sensors-20-02136-t005:** The best average *F*_1_ under various combinations of hyperparameters.

Layer	Kernel Size	Batch Size	Learning Rate	Average *F*_1_	σ	Total Number of Parameters
8	3	50	0.001	69.4	12.5	2,558,340
	5	50	0.001	72.2	12.9	2,687,428
	7	70	0.0005	73.6	11.4	2,816,516
9	3	50	0.001	76.6	9.6	1,608,324
	5	50	0.0001	77.2	9.9	1,868,484
	7	30	0.0005	76.8	11.4	2,128,644
10	3	90	0.0005	77.0	9.4	2,428,292
	5	30	0.0001	77.8	9.1	3,212,740
	7	50	0.001	76.4	10.6	3,997,188
11	3	50	0.0005	77.1	9.9	2,625,412
	5	N/A	N/A	N/A	N/A	N/A
	7	N/A	N/A	N/A	N/A	N/A

**Table 6 sensors-20-02136-t006:** Average *F*_1_ scores of proposed networks and their variants.

Neural Networks	Average *F*_1_ Score	Total Number of Parameters
Proposed-1	77.8	3,212,740
All-BN	69.2	3,216,644
No-BN	76.2	3,212,676
Maxpooling	67.7	2,885,060
Max-Average pooling	75.6	2,885,060
Proposed-2	78.2	3,212,740
Extra-Average	77.4	2,885,060

**Table 7 sensors-20-02136-t007:** Training parameters of the proposed 1D CNN.

Layer Type	Output Shape	Parameters
Conv1D	8996 × 32	192
Batch Normalization	8996 × 32	128
Conv1D	4494 × 32	5152
Conv1D	2243 × 64	10,304
Conv1D	1117 × 64	20,544
Conv1D	554 × 128	41,088
Conv1D	273 × 128	82,048
Conv1D	132 × 256	164,096
Conv1D	62 × 256	327,936
Conv1D	27 × 512	655,872
Conv1D	9 × 512	1,311,232
Dense	128	589,952
Dense	32	4128
Dense	4	132
Total number of network training parameters: 3,212,740		

**Table 8 sensors-20-02136-t008:** Comparison of prediction accuracy of various methods.

Methods	AF (A)	Normal (N)	Noisy (~)	Other (O)
Proposed-1	79.1	90.7	65.3	76.0
Proposed-2	80.8	90.4	66.2	75.3
CRNN	76.4	88.8	64.5	72.6
ResNet-1	65.7	90.2	64.0	69.8
ResNet-2	67.7	88.5	65.6	66.6
CL3-I	76.0	90.1	47.1	75.2

**Table 9 sensors-20-02136-t009:** Comparison of average *F*_1_ score and total number of parameters of various methods.

Methods	Average *F*_1_ Score of A, N, ~, and O	Average *F*_1_ Score of A, N, and O	Total Number of Parameters
Proposed-1	77.8	81.9	3,212,740
Proposed-2	78.2	82.2	3,212,740
CRNN	75.6	79.3	10,149,440
ResNet-1	72.4	75.2	10,466,148
ResNet-2	72.1	74.3	1,219,508
CL3-I	72.1	80.4	206,334

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
