# Peer review of "Detection of Atrial Fibrillation Using 1D Convolutional Neural Network"

_sensors, 2020, doi:10.3390/s20072136_

Round 1

Reviewer 1 Report

Manuscript Number: sensors-745272

Title: Detection of Atrial Fibrillation Using 1-D Convolutional Neural Network
Journal: Sensors.

In this paper, the authors propose an AF detection method based on a 1-D CNN to improve the detection performance while reducing the complexity. The authors used the Physionet challenge 2017 dataset, which includes ECG recordings with different lengths. Due to the requirement of CNN in terms of equal-length recordings, the authors propose a length normalization approach to generate equal-length recordings. Finally, the authors compared the detection performance and the complexity of their proposed method with a few deep-learning-based methods. While the general idea is legit, the paper is not properly written in the format of a journal paper (but rather a class project) and needs significant improvement to transfer it to a proper journal paper.

  • The authors are proposing a multiple segment generation based on 50% overlapping windows for the recordings more than 30 sec. It is not clear how this dependency among the samples can affect their proposed CNN algorithm.
  • While a significant portion of the “introduction” is devoted to the conventional ML techniques (non DL-based), there is no comparison with any of this method in the analysis/result part. I would suggest if the authors are emphasizing on the priority of their DL algorithm compared with other DL algorithms, then reduce the non DL contents from the introduction part.
  • There are redundant explanations about several points including the following.
    • The cross-validation; this approach is pretty standard and is an accepted approach in classification studies. It would be highly recommended to either remove a significant portion of this part or move it to a supplementary document. Further, it is not clear the cross-validation results the authors are reporting in Tables 4 & 5 belong to what hyperparameters.
    • Batch sizes of 10 and 20; although it is understandable that since the optimal batch size (30) falls in the lowest part of their defined batch band, it is recommended to include the extra-small grid search with batch sizes of 10 and 20 in a supplementary doc.
    • Tables 1, 5, and 9 can be moved to a supplementary document as well.

Generally, having an extra analysis/explanations not only does not provide a positive impact on the contents of the paper but also makes the audience more distracted and usually, it is not recommended in scientific papers (unless the authors want to demonstrate some significant benefit they can get from one of the extra measures/explanations).

  • Table 4 does provide an 80:20 F1 score for all four types of recordings. It is expected that this would have similar results to 5-fold results which are contradicting in this table.
  • In section 3.5.2 the authors are talking about the importance of Batch Normalization and Maxpooling methods. However, the results do not show that; it is not clear the purpose of the authors in bringing these much analyses into the paper—again, this significantly distracts the audience form the main purpose of the work.
  • The confusion matrix in figure 5 either needs to show all folds or the average. No specific reason why only the first fold?!
  • It is not clear in table 2, why the Proposed-2 results are different than Average pooling results—it seems that the Proposed-2 method is indeed using Average pooling too—so, it is expected to have similar results which are not the case.
  • The conclusion needs a significant amount of work. Also, there is no discussion on the findings. Usually, at the end of the journal papers, it is expected to have 1-2 pages talking about the general findings of this work and back them up or compare them with the existing literature. This part is totally missing in the paper.
  • While table 11 is a useful table in terms of comparing the detection accuracy and the complexity of the proposed method with the existing methods, first it does not propose any significant amount of improvement in the detection rate (for example in CL3-I average F1 scores of A, N, and O is ~80% which is very close to the Proposed-1 and Proposed-2 results). Second, the complexity does not seem to improve compared with a few of other methods including CL3-I and ResNet-2. It is not clear what exactly are the benefits of the proposed methods compared with the conventional in terms of both detection performance and the complexity.
  • Line 386 is talking about the Max-Average pooling method but then talk only about one Average pooling layer before the flatten layer; it looks like max is missing here.
  • All figures need a significant amount of improvement in the quality to be acceptable for publication.
  • Figure 5 does not have any colorbar label.
  • Figure 6 does not have any x and y axis labels.

Reviewer 2 Report

This manuscript proposes a method for detection of atrial fibrillation in single-lead ECG based on an end-to-end unidimensional convolutional neural networks.The proposed method is very interesting, its development and evaluation is very well structured, and the comparison with the state-of-the-art is complete. Although the results are not ground-breaking, it still manages to offer better performance than the best deep state-of-the-art methods, while requiring less training parameters.

Strengths: (1) good motivation; (2) thorough state-of-the-art review; (3) complete evaluation; (4) better results than the literature; and (5) thoughtful concerns with time constraints.

Weaknesses: (1) some details need further clarification; and (2) class imbalance seems to have been ignored during training.

Main comments

  1. The authors state that the data is single-lead, but never disclose which lead the data is in.
  2. The authors criticise the common data length normalisation methods for being manual. However, selecting a data length threshold using a histogram is still manual, it just is data-driven.
  3. In page 6, lines 224-225, when the authors state “We set 1 to the first place because we want the predicted probability output to be as close as 1”, it is not clear what the authors mean.
  4. The method uses 30s inputs, how would the performance decrease if shorter segments were used? This would be an interesting experiment.
  5. The results show an obvious bias towards the normal label. It is unclear if the authors used any techniques to avoid the effects of the great class imbalance in the data.
  6. In Figure 5, a normalised version of the confusion matrix would be very useful.
  7. Table 11 is very interesting for comparing training time, but what about inference time? The authors should provide a table with average sample inference time for each method.

Minor comments

  1. The authors should disclose the main F1 result in the abstract.
  2. In Figure 2, the authors mention ‘ECG Raw data’ but the data was pre-filtered with a bandpass filter, so this is incorrect.
  3. In section 2.3.2, it seems, at first, that the authors trained the network using classical SGD. Only later I understood that it was, in fact, Adam. The authors should fix this for improved readability.

Reviewer 3 Report

In order to raise the detection accuracy and reduce network complexity, this paper proposes an AF detection method based on an end-to-end 1-D CNN architecture. The validity of the method is verified by the improved data set (Physionet challenge 2017). The results show that, the 1-D CNN method achieves better detection accuracy than the existing DL-based methods. So I think this method is effective. But I have the following suggestions and questions.

  1. Although well-known abbreviations such as "AF" and "CNN" are used, it is recommended to write them in full when they are first listed in the abstract, and abbreviations can be used later.
  2. P1 L38 You’d better change “P wave stands for depolarization” to” P wave stands for atrial depolarization”.
  3. P8 Formula number and formula should be on the same line (L265 266).
  4. It is suggested to introduce the characteristics of data set (Physionet challenge 2017). to highlight the superiority of the algorithm in this paper.
  5. When the algorithm is directly applied to clinical, what are the specific requirements for the data itself.

Round 2

Reviewer 1 Report

Manuscript Number: sensors-745272

Title: Detection of Atrial Fibrillation Using 1-D Convolutional Neural Network
Journal: Sensors.

Most of my comments have been addressed. However, there are a few figures need more improvement:

  • 5 colorbar needs to have a label.
  • 6 y-axis does not have a unit for the amplitude.
  • The font sizes of two top and bottom figures in Fig 8 are not the same: e.g., train/validation/Train History is different between the top and bottom figures. Also, it needs full y-label (e.g., Accuracy).
